# Education for Sustainability: Understanding Processes of Change across Individual, Collective, and System Levels

Elin Pöllänen [1,2], Walter Osika [1,2], Eva Bojner Horwitz [1,2,3] and Christine Wamsler [4,*]

1. Center for Social Sustainability, Department of Neurobiology, Care Science and Society, Karolinska Institute, 17177 Stockholm, Sweden
2. Centre for Psychiatry Research, Department of Clinical Neuroscience, Karolinska Institutet & Stockholm Health Care Services, 17177 Stockholm, Sweden
3. Department of Music, Pedagogy and Society, Royal College of Music, 11591 Stockholm, Sweden
4. Centre for Sustainability Studies (LUCSUS), Lund University, 22100 Lund, Sweden
* Correspondence: christine.wamsler@lucsus.lu.se

**Abstract:** Researchers and practitioners increasingly emphasise the need to complement dominant external, technological approaches with an internal focus to support transformation toward sustainability. However, knowledge on how this internal human dimension can support transformation across individual, collective, and systems levels is limited. Our study addresses this gap. We examined the narratives of participants in the sustainability course "One Year in Transition", using micro-phenomenology and thematic analysis. Our results shed light on the dynamics of inner–outer change and action and the necessary capacities to support them. This related to changes regarding participants' perspectives, which became more relational and interconnected. We also showed that participants increasingly seek an inner space that provides direction and freedom to act. The data suggested that this, over time, leads to increasing internalisation, and the embodiment of a personal identity as a courageous and principled change agent for sustainability. Our results complement extant quantitative research in the field by offering a nuanced picture of the entangled nature of inner–outer transformation processes and associated influencing factors. In addition, they point towards ways in which inner dimensions can be leveraged to achieve change, thus filling existing knowledge gaps for reaching sustainability and associated goals across all levels.

**Keywords:** inner transition; sustainability; subjectivity; education for sustainability; transformative education; embodied knowledge; micro-phenomenology

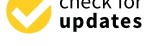



## 1. Introduction

The issue of climate change is posing an increasing challenge to sustainable development worldwide [1–4]. Our current policy approaches have not been able to address the magnitude and rate of change that is needed, and it is increasingly clear that sustainability goals and targets that have been established at international and national levels will not be reached [2,5]). Sweden is no exception to this, even though the country is often declared a forerunner in climate governance.

The way in which climate change and related approaches have been framed in Sweden, and worldwide, has been closely related to the biophysical discourse in which climate change has mainly been seen as an external technical challenge [6] (In this article, external and internal are used synonymously as outer and inner. Transformation and change are used interchangeably and relate to the same idea of a fundamental change in a system (see Ref. [7]). Consequently, much focus has, so far, been placed on solutions that address external socio-economic structures and technology [8]. It is, however, becoming increasingly clear that such approaches alone will not meet the 1.5–2 °C climate mitigation target, because the current mechanistic stance is not tackling the root causes of the problem.

Scholars and practitioners are thus increasingly emphasising the need to complement the current external perspective with an internal focus identifying individual and collective beliefs, values, worldviews and associated inner (cognitive, emotional, relational) capacities [8–10], something that has also been highlighted in the recent reports of the International Panel on Climate Change [11,12]. This type of transformation involves exploring the role of the internal dimensions of human life, and the links between climate change and society. It includes connections with other pressing societal crises, such as health inequalities, and other forms of injustice [6,13,14]. Accordingly, scholars have increasingly pointed out that only focusing on direct drivers, or so-called 'shallow' leverage points (e.g., technological innovations) rather than indirect, or so-called 'deep' leverage points (mindsets, values, norms), to reach mitigation targets could not only hinder progress, but even have negative effects on sustainability goals, through overlooking important synergies [5].

In this emerging rationale, climate change and other societal challenges can be partly understood as manifestations of our internal mental states: we enact our inner mindsets, value systems, beliefs, and worldviews [10,15,16]. The substantial role that internal dimensions play in shaping social life is supported by several theories, notably systems theory [17–19], integral theory [20], the three spheres of transformation framework [21], Theory U [22], and the internal–external transformation model [10]). They relate to behaviour change theories and social models of behaviour change that bring together structural and individual factors e.g., social practice theory [23–25] and sustainable transition management [26,27] but promote a more relational understanding of sustainability transformation [28,29]. Particularly, they depict the potential of all individuals as agents of change for sustainability across individual, collective, and system levels (ibid). This article focused on the so-called inner sphere of transformation (related to internal individual and collective change and its influence on sustainability across scales), which means that the entry point is not on the practical or political spheres of transformation (i.e., behavioural change and systems change).

Emerging concepts and terms, such as ecoanxiety, ecological grief, solastalgia, and biospheric concern are another indication of the growing interest and research into the internal implications of external events, and vice versa [30–34]. Experiencing extreme events first-hand, or through feelings of uncertainty and various forms of media can, for instance, have short-term and indirect long-term health implications that impair our capacities to act at an individual and societal level, and which further increase climate risk [31,35]. Transformative efforts are thus needed to safeguard both individual and planetary health and wellbeing.

Although the links between internal and external system change for sustainability are receiving increased attention, there are important knowledge gaps concerning the underlying dynamics and processes that are essential for inner change, and how these translate into impactful action across scales [6,10,13,36–38]). More empirical research is needed to understand how certain internal human dimensions, or states, accelerate transformation towards sustainability (see e.g., Refs. [8,10,13].

Accordingly, scholars are now more and more often challenging sustainability education and the competencies that are needed, as they have so far mainly been discussed from a cognitive perspective and disembodied and detached from emotions, arguing for the neglected but important role of inner dimensions in sustainability work [13,39–42]. Various competency frameworks have recently emerged to support sustainable development, but it is only recently that they have also recognised the importance of inner capacities, and related empirical work is vastly lacking [43,44]. Educational settings still impose the risk of hampering transformative learning due to their ontological and epistemological traditions, where ideas of compartmentalisation and reductionism fail to capture the complexity and interconnection of sustainability issues [13]. As a response, courses and educational programs are offered with aims to offer a more holistic and relational worldview with different modes of knowledge [28,29,38] that also tap into the learner's inner perspectives and bridge the gap between theory and practice [39,40]. For instance, UNESCO has recognised the

importance of education that goes beyond transferring scientific information concerning environmental issues and that includes interdisciplinary pedagogy that empowers learners and enables self-direction in the learning process [45].

Our study addresses a research gap as it aims to analyse inner–outer change processes at individual and group levels that have the potential, in turn, to support or catalyse systemic engagement for sustainability. We examined the adult education program called One Year in Transition (1YT), an international program situated in Sweden. This course was chosen because of its declared aim to address sustainable transformation by linking inner and outer dimensions, and by combining solutions at individual, collective and system levels. The program consists of four modules, spread out over a year, which introduce concepts that include sustainability, inner transformation and transition, permaculture, integral theory, and Theory U. It combines study visits, online lectures, peer support groups and mentoring, as well as contemplative practices [22,46]. The program starts with building trust, providing an overview of current sustainability problems, and introducing the concept of inner transition and transformation [8,9]. The course literature includes Joanna Macys "The work that reconnects", to deepen the understanding of what she has named the "Great turning" [46], which involves the emergence of new and creative individual and collective human responses. A related contemplative practice is the 4.6 km "Deep Time Walk", where every meter represents a million years of planetary time. A core feature of the program is open dialogues between participants on the possible future and paradigm shift. In addition, participants are expected to develop and run their own sustainability projects within a specific context and/or local community. More information on the One Year in Transition course is available at: https://eskilstunafolkhogskola.nu/distanskurser-helgkurser/ett-ar-i-omstallning/ (in Swedish) (accessed on 10 January 2023). The course took its inspiration from the Transition Network: https://transitionnetwork.org/news/one-year-in-transition-1yt/ (accessed on 10 January 2023).

## 2. Methodology

In accordance with the problem and theories outlined above, and the aim of our research, we chose to apply a narrative methodology with a micro-phenomenology base [47–51]. Our study did not follow a conventional case-study approach in environmental and sustainability education, as this would not have been adequate given our research focus (see Ref. [52]). Micro-phenomenology (MP) research shows that we can learn to describe fine lived experiences very reliably by asking questions such as "how do you know that you feel sad?" or "how do you know this is important for you?" [49–51]. The MP methodology enabled us in this study to access participants' reflective minds based on a two-step approach: (1) an individual in-depth interview and (2) a reflective group-based discussion. The texts from the interviews were analysed through a thematic analysis [53,54], using a reflexive method [55]. The aim was to understand experience from a first-person perspective, through facilitating respondents' access to their inner dimensions and reflective minds [56–58]. Participants make sense of complex situations through the identification of underlying patterns [59–61]. Such everyday micro-narratives reflect the meanings that people assign to the world, and therefore give access to an everyday form of social knowledge, extracted from rhetoric, which can help to reveal elements of a grounded discourse that informs people's decisions and, ultimately, their actions [62].

### 2.1. Data Collection

Data collection included one single participant's first-person perspectives from the course and one group discussion (involving all course participants who were available, totalling 12), to be able to capture both individual processes and dimensions and group dynamics. Participants applied to the course themselves, were between 18–75 years, and came from geographically diverse areas in Sweden. Their occupational backgrounds were diverse, including social work, management, human resources, handicraft, farming, and environmental studies and leadership. Due to COVID-19 restrictions, data collection took

place remotely, using Zoom. Both the interview and the group discussion were recorded and transcribed; the same process was followed, and the same set of questions was used. The latter were based on a theoretical understanding of the topic, drawing upon prior research, and adopted a narrative and micro-phenomenological combined approach (see Ref. [57]). In order to understand how the 1YT program had shaped participants' internal dimensions and change processes, the interview and discussion started with sharing their first-person experiences. This led to questions that explored underlying processes and embodied capacities in relation to internal and external change. Thus, individual narratives contributed to the depth of the group narrative.

*2.2. Data Analysis*

We examined participants' narratives with thematic analysis, which takes a first-person perspective when interpreting narrative data. This interpretative approach helps to identify and analyse pattens and themes in the data. It goes beyond a semantic analysis, as it identifies underlying patterns and assumptions that shape the semantic content found in the data [53,55].

Consistent with Braun and Clarke [53], and Nowell and colleagues [54], we adopted the following six-step approach found in thematic analysis: (1) familiarisation with the data; (2) generating initial ideas and themes through open coding; (3) interpreting and systematically categorising the content of interview transcripts into themes and associated patterns; (4) reviewing; and (5) further defining through axial and selective coding. The aim of the final step was to produce descriptions from themes. It is important to highlight that these steps are part of a circular (and not a linear) process as the analysis moves forward and backward across the data.

Two researchers, who were not present during the interviews, initially analysed the material independently. Interpretations were discussed between the two researchers after initial familiarisation and coding. By examining keywords and the linkages between them, distinct categories were identified that led to the identification of overarching themes. Whilst agreement and data saturation were achieved, this was not the main aim, but rather, to familiarise further with the data and explore other interpretations of the data and assumptions made by the coders. Identified themes were also presented and discussed with all the study's researchers with the same aim. Themes were identified using an inductive approach rather than pre-existing theoretical frameworks and were the endpoint of the analytical process, found in the intersection of the dataset and researchers' understanding and interpretation.

A crucial, and classical question within the thematic analysis method is: what is considered to be a theme [53–55]? Answering that question can be difficult, especially for topics where the subject matter is interrelated. The stance taken in this research was to identify themes that captured aspects that directly related to the research aim, and represented meanings or patterns found in the overall dataset. A theme could re-occur in the dataset, but more importantly, had a profound effect on the narrative/broader context. Thus, the final themes carry forward stories of shared meaning, whilst including individual narratives and perspectives (see Tables 1–4).

**Table 1.** Theme 1–Relational and interconnected perspectives.

| Raw Data–Illustrative Citations | Quote Summary | Sub-Themes |
|---|---|---|
| "To me, the biggest realisation has been that everything must start from inside, with inner transformation and personal development, before too much focus is placed on outer manifestations." | Realisation of the importance of inner dimensions and transformative processes for sustainable (long-lasting) outer manifestations and action. | Shift towards including inner dimensions for achieving sustainable change. |
| "This year, I have in some way understood that it is about shifting mindset and relationships to each other and nature, to a more relationship-based thinking." | Seeing the necessity of addressing mindsets/our relationships to others and nature for achieving transformation. | Shift in seeing sustainability challenges as relational/relationship problems. |
| "For several years, I have planted vegetables and thought about how I can make it on my own. But [over time the question has shifted to] how can I cooperate in my local community, with my neighbours [to achieve change]?" | Seeing the necessity to link to the wider community, and seeing one's role as part of a wider, interdependent system. | Shift towards increased cooperation, and collective change-making. |
| "Everything in nature and society is dependent on each other and belongs together. To value things for what they are and not for what value they can give us [has been an important realisation]." | Feeling interconnection and interdependence, and valuing things for what they are, rather than what they can give us. | Non-instrumental/mechanical, intrinsic values are seen as a driving force for engaging in sustainability and transformation. |

**Table 2.** Theme 2–Inner space-making, freedom and creation.

| Raw Data–Illustrative Citations | Quote Summary | Sub-Themes |
|---|---|---|
| "In the beginning, I thought this course would be something that I just did on the side for some inspiration, pretty shallow. But it has truly gone into the deepest of my inner self ( . . . ). I let go of some preconceived ideas of how the world is, and what is possible for me. One of the most challenging things have been to challenge these comfortable patterns and behaviours, but it has also been freeing." | Exploration of preconceived beliefs and worldviews through finding an inner space for reflection and action that is freeing. | Inner space-making as a source of self-reflection and action. |
| "Creativity is a way of uniting, to touch one another. For example, when you sit next to someone at a concert and you have no idea who the person is, but you have that shared experience, of being moved." | Importance of creativity and shared experience to increase togetherness, and act from this space. | Creativity and co-creation. |
| "[being moved] perhaps means that things are re-arranged, dissolve, or lose their structure for a while, and that somehow makes it possible, or sometimes necessary, to make other choices." | The importance of creating an inner space to be vulnerable, being moved emotionally, losing or rearranging current perspectives, and opening up space for new, different choices and actions. | Inner space as an opening for vulnerability and change. |
| "It is like a liberation. I feel free. I feel boundless, I feel like a creator." | Linking inner and outer aspects of change and having the opportunity to manifest them feels freeing and empowering, a kind of freedom to act, despite (or due to) boundaries. | Sense of inner freedom and empowerment. |

**Table 3.** Theme 3–Embodied transformation–individual and collective awareness and growth.

| Raw Data–Illustrative Citations | Quote Summary | Sub-Themes |
|---|---|---|
| "When something feels right, then I feel a calmness in my stomach, like a heaviness . . . During the course, I have managed to put that feeling into words, feel it, and become aware of what it is; a confirmation coming from the inside." | Integration of bodily sensations into understanding and judgement, e.g., when something feels 'right' or 'wrong'. | Increased awareness of bodily sensations and emotions, and their interconnectedness. |
| "When I get inspired by something, then I get very happy, and I think that energy shows. In that way, I can make more people join, and I think that could be a good way forward." | Embodiment as a form of energy, linked to interconnectedness and change-making. | Importance of the body–of recreation connection–in change-making. |
| "I am very much in my head when I am in my own intelligence, I feel it very clearly. But when it's a collective intelligence, I feel it more in my heart, around the heart area, the whole body is more involved. Or it's more coming from the middle of the body . . . " | Awareness of collective intelligence as altered mind-body topography, compared with individual intelligence. | A collective transformation with an embodied group culture. |

**Table 4.** Theme 4–Lived embodiment–lived embodied experiences.

| Raw Data–Illustrative Examples | Quote Summary | Sub-Themes |
|---|---|---|
| "Life has become the course." | There is no difference, separation or disconnection between inner and outer knowledge, and between inner and outer change. | Overcoming separation/disconnection. Reconnection and oneness. |
| "I have read academic courses [ . . . ] that have awakened curiosity within me, but it has stopped there, and has never connected with this deeper dimension." | Learning is connecting with a deeper exploration of self and personal growth. | Embodied learning. Vertical learning. |
| "[The inner compass] has always been with me, but it has gotten more space to grow [ . . . ] now at work, I feel that I have to be the voice of discomfort." | Reflection and support of the inner compass, and its embodiment to support change. | Embodied change–leading to collective and systems change. |
| "I have become convinced that I have to deal with the worry of others . . . " | Seeing the importance of the embodied emotions of others. | Mirrored embodied transformation to support collective and systems change. |
| "[It] moved me very much [ . . . ], leading to different 'conversations' and actions, including an article that was published in a newspaper." | Importance of emotions for action-taking. | Embodiment as the alignment of emotions, values and actions at individual and collective levels. |

## 3. Results

Our analyses led to the identification of four interrelated themes. They involve changes regarding respondents' relational perspectives (Section 3.1), and the finding of an interior space or stillness of self-awareness and reflection, from which a kind of freedom, integrity, and creativity for taking action emerges (Section 3.2). From this inner space, an increasing (sense of) embodiment of inner–outer change (Section 3.3) develops over time. It manifests in actions for sustainability, by allowing people to 'stay with the trouble' through courageous principled action, taken at individual and societal levels, from a place of meaning and purpose (Section 3.4).

### 3.1. Relational and Interconnected Perspectives

The dominant theme concerned the expression of a change in respondents' perspectives towards more relational and interconnected views of the world, and one's role within it. It denotes a shift in respondents' beliefs and worldviews, and associated values.

In the individual and group interviews, participants described experiences that demonstrate mindset change towards a more interrelated, layered view of reality. This became particularly apparent in the use of non-dualistic (e.g., not either/or) language regarding individual, collective, and systemic processes. An example is the Nature–Culture dualism prominent in Western thought, which can be considered an obstacle for a more integrated worldview for sustainability (see e.g., Refs. [63,64]). At the individual level, all participants

stated that the 1YT program had made them more aware of the role of the mind (emotions, thoughts), and the mind-body interplay in everyday life and transformation for sustainability. At the collective level, most participants noted an increased sense of connection and collaboration for transformation and change-making and being on a quest for more relation-based thinking, being, and acting. At the broader system level, participants also expressed the broader importance of relationships, not only to other humans, but also to non-human beings and nature. The striving for a more relational form of thinking seems to go hand-in-hand with participants placing more intrinsic (rather than instrumental) value on other humans and nature, and their own inner development (see Table 1). The latter also relates to a change in feelings of resilience, stemming from a feeling of connectedness or oneness of life.

The identified change in relational and interconnected perspectives was also linked to individual agency, and engagement in sustainable action. For instance, one participant stated that there could be a link between "personal energy saving" and "outer energy saving", where increased awareness and setting boundaries could help to generate well-being, a feeling of purpose and, ultimately, increased freedom to act (see Section 3.2). Accordingly, learning about one's own inner dimensions was not only seen as crucial for outer transformation, but also as having a clear value of its own. In this context, participants highlighted the importance of action not merely being results-driven, but as being part of a relational process they are going through, which includes multiple and intertwined inner and outer change processes. Finally, participants noted that gratitude and humanization—being "human and not a machine"—were important features that had increased their appreciation of themselves, others, and their surroundings.

Relation-based thinking was expressed as a catalyst for taking action, identifiable in a shift towards cooperation rather than competition/performance. It was, amongst other things, expressed in participants' search to understand their place in the world, whilst increasing their circles of identity, care, and responsibility. Notably, they sought to identify areas where they could catalyse change in relation to their direct surroundings, such as among family members, neighbours, the local community, and wider society (see example in Table 1).

### 3.2. Inner Space-Making, Freedom and Creation

The second theme is directly related to the first, as it links to the described shift in beliefs, worldviews, and values. It is attributed to the described change in one's inner space of being. The latter influences a sense of freedom, out of which a different kind of action and 'creation' emerges. It differs from the first theme in that it is more oriented toward action.

Participants emphasised the importance of self-reflection, opening up, and becoming vulnerable in order to tap into an inner space out of which change, creativity, and co-creation emerged. A safe container for speaking openly about inner dimensions, and how they interrelate with outer transformation at the personal and group levels, was seen as a pre-condition for growing together, and creating an inner space that enabled personal growth, being moved, and acting from within. The transformation was expressed as starting from within, then growing into a collaborative, collective presence that participants had not felt or experienced when the course started.

A dominant sub-theme was the feeling of freedom and liberty that emerged through a process of rearrangement. The latter was described as becoming increasingly aware of one's inner reactions and processes and acting more in unison with one's intrinsic values. Many participants found themselves feeling increasingly empowered and liberated, with the practice of sustainability being part of the process. This sense of liberty, and the joy of creation were seen as important elements in participants' relations to transformation and sustainability, where outer manifestations (actions or inaction) directly stem from the boundaries, limitations, and actions perceived to be vital for systemic change. The analysis of data from individual interviews, and the group discussion underlined that participants

sought to construct a connection between individual and collective freedom, and a society that could live within the limits of the planet's resources.

### 3.3. Embodied Transformation–Individual and Collective Awareness and Growth

The third theme relates to changes in perspectives that manifested over time, in an increasing (sense of) embodiment of inner–outer change.

Participants expressed an evolving understanding of inner processes and bodily sensations and connected this understanding to inner transformation and outer change-making. Moreover, they described the importance of being "in the whole body" with one's "whole energy" and being able to identify and relate better to their own biases and reactions (e.g., emotions, fears, and other triggers). Being in one's body was seen as an important aspect of knowing, along with being able to express oneself with one's whole being, being authentic, and, thus, inspiring others.

During the group discussion, participants described bodily manifestations not only at individual, but also at group level, and in great detail. For instance, they noted an increased awareness of internal processing and external manifestation, and transformations of these processes, also related to the group as an embodied whole. The group was described as a 'collective body', which had come into existence during the course, and was continuously changing. For instance, participants observed the presence of both individual and collective voices, which had changed over time to become lower in frequency, and calmer.

### 3.4. Lived Embodiment–Change as Being–Being Change

The final theme relates to the embodiment of inner–outer change, where actions can be seen as a manifestation of change as being. It is related to the previous theme but has a stronger orientation toward (embodied) action. (The theories concerning embodiment refer to the role of the body in making sense of our thoughts and feelings, meaning that our bodily awareness has an impact on our cognition; how we understand ourselves as well as the world around us (see e.g., Refs. [65–67]).)

A clear expression of this process of change as being was the statement that "life has become the course." Transformation was not merely seen as taking in information, but about creating space and opportunities for self-reflection, and tapping into one's inner being (see Sections 3.1–3.3). This supported lived and embodied experiences, and catalysed action through creating personal insights, and strengthening an inner compass (see Section 3.2).

Embodiment was seen as being able to "stay with the trouble" of climate change, and associated inner dimensions and emotions, through courageous, principled, mirrored action-taking at individual and societal levels, from a place of meaning and purpose. In fact, daring to stay in discomfort was part of the expressed changes, and was seen as a key attribute to address sustainability challenges (see Table 4). Participants described using discomfort as a guide throughout the course and noted that it helped them to cultivate the capacity to discern situations where it is important to act in line with one's values. They underlined the realisation that discomfort often comes from fear, anxiety, social norms, or certain biases. As described by one participant, it can be overwhelming to care, which, in turn can result in disempowerment, apathy, and feeling hopeless. At the same time, discomfort was seen as an important gateway to moving towards action-taking. Participants described how they had learned to stay strong in discomfort, even when their social context marginalised or explicitly opposed sustainable ideas and visions, for instance in some workplaces. In addition to the enabling factors mentioned above, respondents emphasised the importance of having access to a learning environment where failure is both accepted and seen as a part of the learning process.

## 4. Discussion

Our findings suggest that the 1YT program supports and catalyses actions for sustainability. It engages participants in a process which supports certain inner capacities that, in turn, change (or strengthen) certain worldviews, beliefs and values, reflected in

how they perceive their own being, thinking, and acting. The identified changes indicate a move towards relational and interconnected perspectives, and finding an interior space, or stillness, for self-reflection. From this, freedom, integrity and creativity for taking action emerge, which ultimately manifests in a (sense of) embodiment of inner–outer change that allows the individual to "stay with the trouble". These outcomes confirm recently developed models of internal–external transformation [10] and complement them by providing a nuanced understanding of the intertwined nature of the underlying inner–outer transformation processes, and related influencing factors.

Participants described a reciprocal relationship between individual and collective transformation, as well as between inner and outer processes, which is in line with increasing calls for more relational approaches to sustainability [28,29,68–72]. Accordingly, our findings are in line with, and illustrate, in granular empirical detail, theoretical notions of sustainable transformation and associated leverage points [17,18].

In addition, the inner aspects that we have identified as crucial for transformation support recent knowledge development and theory on transformative capacities or so-called inner development goals [73]. These capacities can be grouped into five clusters: *awareness*, *connection*, *insight*, *purpose*, and *agency* [10,37], which are all reflected in the identified themes (see Sections 3.1–3.4). In addition, our analyses of the empirical data illustrate how the necessary capacities interrelate and translate into internal and external change. *Insight* is, for instance, visible in participants' changed use of language, which becomes non-dualistic and multi-layered, and their increased focus on relations, perspective-taking, and the integration of different types of knowledge. This was, in turn, closely related to changes in self-*awareness*, a realisation of one's potential influence and, ultimately, one's intrinsic motivation or *purpose*.

*Connection* was one of the most dominant factors. It is linked to, and has implications for, all the themes and associated capacities identified in our data. Feelings of connection to self, others, and the environment were strongly related to reflections and insight on purpose, in other words, the activation and reflectivity of one's values, intentions and responsibility led to an increasing circle of identity, care, and responsibility (see Table 1). The importance of connection has been underscored in sustainability (see Ref. [10]), leadership development [74] and theories concerning education and adult development [75]. The interviews highlighted an intrinsic value orientation. Inner and outer processes, along with living beings and natural systems, were ascribed internal value, rather than just being valued for their function (see also the theory of mind described in Frith and Frith [76]). The perceived oneness of life was not necessarily seen as a unification, but rather as tuning in to the individual's surroundings (portrayed as, for example, "wave motions around bodies of relation") as a source of action-taking. Previous research has for instance found a sense of connection and relatedness to nature to predict pro-environmental behaviours [77], whereas social identification and connection to animals has been linked to a greater desire to help animals in an altruistic and empowering way [78].

Many participants described increased *awareness*. They highlighted how they had experienced increased presence, attention, and emotional processing that had enabled increased psychological and cognitive flexibility. *Awareness* also relates to strengthening capacities to listen and communicate and being open to change [10]. This is confirmed by our study, where participants stated that they had developed a toolbox that helped them to speak about, and understand their own reactions, emotions, and impressions. The provision of a space where they could be vulnerable and safe, and visualise and practise the manifestation of inner–outer transformation, enabled them to continue 'listening inwards', to be moved, to engage, and consequently to rearrange, dissolve or restructure their inner and outer lives (see Tables 2 and 3). In addition, in line with an increasing recognition that creativity has an important role in tackling sustainability issues across sectors at multiple levels [79–82], participants saw creativity as an enabler for co-creation, and empowerment to transform in a way that moves beyond the *status quo*.

Importantly, being able to *practise agency* supported participants' sense of empowerment and inner capacities that enhanced collaboration. Creativity played an important role in the experience of feeling free from certain limitations, and the joy of being an active (co-)creator of change (see Table 2). As illustrated by all four themes, transformation was achieved by enabling participants to tap into their inner potential, interests, and abilities. This finding is supported by previous studies, which have found that a culture of creativity and learning empowers individuals to take action to materialise their inner vision and values [79,83]. Participants described their transformation narrative and experience in relation to different timeframes, reflecting specific moments, or shorter or longer periods of time where transformation and associated inner capacities evolved or emerged. For some, certain life-changing events that had occurred before the course had forced them to think about changing their path. A phenomenon that has been described in individuals with, for example, exhaustion/burnout symptoms [84]. At the same time, all participants were open to the idea of transformation and, thus, were interested in taking the course in the first place. Our analyses show that the group engaged in a collective experience which developed throughout the course, and that is interrelated with, but differs from each individual's experience over time. Independent of time or context, our analyses illustrate the role of self, both as a way to prompt inner reflection, and as a potential entry point for interventions to spur collective change, as indicated by previous research [13,85].

Furthermore, our analyses show that the transformation of our inner selves to support sustainability entails being able to deal with deeply embodied emotions; this poses a significant challenge and relates to all five clusters of transformative capacities [10]. As with any change, transformation is closely related to individual and societal resilience, or the ability to utilise inner strengths and outer resources [86]. In this study, many participants explicitly acknowledged the importance of emotions and emotional processing for change, decision-making, and action. Using emotions as a tool, and discomfort as a guide were particularly emphasised (see Section 3.4). As the course progressed, participants developed a deeper understanding of their inner feelings and processes in relation to their surroundings, and how to act accordingly. This outcome resonates with studies suggesting a link between mindfulness and sustainable climate adaptation and non-fatalistic attitudes [87,88], as well as current theories regarding embodiment, where all experiences are, ultimately, embodied, and knowledge is accessed through the senses [66,67]. Current theories on learning show that emotions, bodily sensations, and cognition are intertwined, and underlying motivations and intentions are crucial [89].

In addition, the idea of not shying away from discomfort but rather placing it at the centre of transformation united several themes (see Section 3.2 especially regarding vulnerability, Section 3.3 on bodily sensations/triggers and Section 3.4 on embodiment). The role of discomfort was present in participants' language and was closely tied to notions of creativity and change-making. By identifying the role of different kinds of discomfort, as well as accepting discomfort during change, participants developed the ability to differentiate between when to move away from discomfort or move towards/stay in it to support individual, group, and societal sustainability. Using terminology from relational frame theory, this can be interpreted as an improvement in participants' psychological flexibility during the course, suggesting that psychological flexibility is an important mechanism of change [90,91]. This observation is also in line with studies that show that experiential avoidance, and associated cognitive entanglement are key features in psychological inflexibility. The latter can be seen as the cause of both resistance to needed transformation, and psychological suffering [56]. It can thus be hypothesised that this understanding has translated into an acceptance amongst participants, despite its impact on the individual's sense of discomfort.

Creating safe spaces that allow participants to engage in feelings of discomfort and reflexivity seem to be critical in setting the context for challenging individual and collective paradigms that underlie sustainability challenges [37]. There is also the suggestion of how transformative pedagogy can make use of discomfort, criticality, and uncertainty in

teaching and thereby create "brave spaces" (see Refs [91,92]). Moreover, blaming others and common us-versus-them dynamics only seem to perpetuate unsustainable systems instead of exploring underlying feelings of anxiety, greed or guilt [28,29,93,94]. In line with this, other studies increasingly highlight that ecological crises such as climate change originate in our 'blindness' as to how they relate to our individual and collective exploitative mindsets [16,58]. This blindness stems from failing to take into account the fact that human awareness is not only partial and biased, but also results in inadequate climate responses and conditions in which we are exhausted and are exhausting the Earth [16]. To regain contact with our real experience, scholars thus increasingly argue that we must end the transformation of all aspects of our lives into an object of consumption, whilst developing the courage to change our society across individual, collective and system levels [10,58]. Participants echoed this observation, for instance pointing out how they had started treating themselves more as humans rather than machines, with a focus on finding a balance between engagement and rest, which they linked to concepts of de-growth and societal sustainability.

Overall, our analyses show that the 1YT program catalysed inner individual processes and had an influence that goes beyond individual lifestyle changes. Through strengthening psychological and cognitive flexibility, emotional regulation, and establishing the conditions and processes for creativity and trust to link inner and outer transformation, sustainability became an embodied way to save energy and regenerate at personal, group, and societal levels. The combination of tools and practices decreased feelings of overwhelm, apathy, anxiety, and fear. Together with the creation of an environment of learning, where mistakes were invited and non-stigmatized, the threshold for action-taking was lowered. Our study thus contributes to the growing field of education for sustainability and associated calls for more relational and integral approaches and pedagogies that support inner–outer change [5,13,28,37,38,45,57,60,95–97].

At the same time, it is important to highlight some of the limitations of our research. The notion of time could have been examined in more detail. Pre-assessments and follow-up work could help to track changes amongst participants and contribute to drawing some conclusions concerning temporal aspects of transformation, environmental engagement, and the impact of education and inner–outer transformation processes over time. Although some participants shared concrete actions at collective and systems levels as a direct result of the course, the study did not investigate examples of direct action or habits, or include follow-up of such actions in order to assess potential under or overestimates, notably of social desirability bias [98]. As participants in the study were self-selected and volunteered to participate, we cannot draw any conclusions regarding inner–outer transformation processes in the general Swedish population. In addition, since participants applied to the course, they have a general interest in environmental issues and related transformation. Other limitations relate to the challenges encountered during the interviews. Technical obstacles and tools (e.g., the use of Zoom) shaped the sample, i.e., the willingness of the course participants to take part in the study, how the methods were executed, and how data could be interpreted. Specifically, it was difficult to identify who was the leader of certain narratives and to distinguish different voices within the group. However, such challenges were overcome by the fact that the interviewer was very experienced, and trained in qualitative interviewing and micro-phenomenology, and any potential doubts could be checked through follow-up questions. Finally, whilst the reflexive approach leaves room for interpretation and subjectivity, it is important to highlight that, in contrast to representative data and larger samples, it has an important role as it can uncover underlying processes and patterns that cannot be accessed with quantitative approaches [99].

## 5. Conclusions and Future Directions

Our study shows that transformation is a multi-layered, complex process where inner and outer processes develop simultaneously, and interact. It also involves change to our inner dimensions, behaviours, cultures, and systems. The role of education in

transformation lies, in this context, not only in terms of knowledge development, but also in its potential to create and hold spaces in which individuals can experience, reflect, and begin to embody personal insights. This helps to relax our dominant individual and societal beliefs, and worldviews, and develop the inner capacities and space from which to take action. Nourishing capacities and intrinsic values ultimately offer greater promise for practice, by manifesting this change in mindsets, values, and norms both individually and collectively. Furthermore, engagement in such experiences leads to the creation of communities of practice, which may also help to deal with the discomfort of "staying in the trouble", and bolster courage to continue embodying these practices outside of one's direct circle. Methodologically, our study shows that micro-phenomenology can help shed light on how spaces of inner transformation are emerging and translate into outer change. Micro-phenomenology methodology helps guiding participants towards deeper reflections, associated emotions and thoughts, whilst also the interviewer accounts become more in-depth.

Seeing inner–outer transformation as a practice and on-going process, rather than a discrete event or an end in itself, helps us to understand that it must be nurtured, and how this might be done. This represents an important step forward in supporting individual and planetary wellbeing; a step that requires dedicated support and consideration regarding how it is integrated into sustainability education, along with practice and policy that aims to address the sustainable development goals. With this in mind, the following areas will be important for future research and policy development.

First, we need to better-understand how to integrate different disciplines and methods, in order to nourish inner–outer change at individual and collective levels. It would, for instance, be valuable to integrate the study of positive personality change, and the known stages of personal development, in order to identify periods that offer greatest receptivity to change. This insight would feed into pedagogical design in sustainability educational settings [10,100]. Accompanying this, we need to increase our understanding of which methods and processes can best-tap into inner experiences, including subtle changes and tipping points that translate into the likelihood of action [37,57,58,101–103]. Such findings would inform the (re-)design of existing educational programs or sustainability courses within curriculums and recommend appropriate adjustments. These changes could include accessibility, modes of learning, and methods that promote a greater sense of interconnection and belonging. Importantly, longitudinal studies are needed to explore the effects of educational efforts that address inner dimensions. This may involve tracking participants' connectedness and attitudes to nature, people, and animals over time, as well as their willingness to make personal sacrifices and trade-offs to the benefit of, for example, ecosystem health, social justice and equity, and animal rights and welfare [104].

Second, we need to better-understand the integration of knowledge systems and the decolonization of education. Going forward, our research and actions should be informed by the knowledge of indigenous groups about environmental sustainability that has existed, but remained marginalised, for hundreds of years, and is intrinsically linked with the identified themes of inner–outer change [10]. In addition, the rights and interests of these groups must be brought to the fore.

Thirdly, future research could explore the role of narratives and new imaginaries for inner–outer change. For example, it would be relevant to look further into the relationship between change in narrative identity and adversity, and the educational setting, or links between change in narrative identity and manifested behaviours towards transformation for sustainability [105–108].

Finally, research cannot neglect a deep analysis of the political dynamics at play in particular contexts. For example, few studies have focused on workplace and professional settings, in which the dynamics of, and resistance to change vary. As with other contexts, research is needed to expand related knowledge by exploring the link between inner dimensions and policymaking at the national or the international level, and between inner dimensions and the processes of innovation and problem-solving, whether social or

technical. It is only by having a holistic picture of the ways in which inner dimensions shape transformation in an external sphere that we can hope to achieve success through these external interventions. This is key for both personal and planetary wellbeing, particularly in order to meet the sustainable development goals and implement climate policies at local, national, and international levels.

**Author Contributions:** Conceptualization, C.W., W.O. and E.B.H.; methodology, E.B.H., E.P., W.O. and C.W.; validation, E.P., W.O., C.W. and E.B.H.; formal analysis, E.P. and W.O.; investigation, E.B.H.; resources, C.W. and W.O.; data curation, C.W. and E.P.; writing—original draft preparation, E.P.; writing—review and editing, E.P., C.W., W.O. and E.B.H.; visualization, C.W., E.B.H. and W.O.; supervision, W.O., E.B.H. and C.W.; project administration, C.W.; funding acquisition, C.W. and W.O. All authors have read and agreed to the published version of the manuscript.

**Funding:** The research was supported by two projects funded by the Swedish Research Council Formas: (i) Mind4Change (grant number 2019-00390; full title: Agents of Change: Mind, Cognitive Bias and Decision-Making in a Context of Social and Climate Change), and (ii) TransVision (grant number 2019-01969; full title: Transition Visions: Coupling Society, Well-being and Energy Systems for Transitioning to a Fossil-free Society).

**Data Availability Statement:** Restrictions apply to the availability of these data. The data is not publicly available due to privacy and ethical considerations.

**Acknowledgments:** We would like to express our appreciation to the 1YT course participants for making this study possible and, in particular to Peter Hagerrot for supporting the project by communicating with participants, observing the process, and patiently answering our questions.

**Conflicts of Interest:** The authors report there are no competing interest to declare.

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
