# Peer review of "Education for Sustainability: Understanding Processes of Change across Individual, Collective, and System Levels"

_challenges, doi:10.3390/challe14010005_

Round 1

Reviewer 1 Report

This is a very interesting study that is very important to helping create curricula around climate change action by examining the mental processes and mindshift changes that take place throughout the course. I think one significant improvement would be to have tracked the changes in beliefs from the beginning to the end of the course more intentionally. Other than that, there are a few typographical errors, but everything else is acceptable.

Author Response

Comments

Responses

1

OVERALL COMMENT & ASSESSMENT

“This is a very interesting study that is very important to helping create curricula around climate change action by examining the mental processes and mindshift changes that take place throughout the course.”

We thank the reviewer for the positive and valuable feedback and for highlighting the importance and relevance of our research.

We have aimed to adjust the manuscript in line with the reviewer’s comments.

2

“I think one significant improvement would be to have tracked the changes in beliefs from the beginning to the end of the course more intentionally. Other than that, there are a few typographical errors, but everything else is acceptable.”

Based on the reviewer’s comment, we have now revised the article to improve typographical errors in the text.

Whilst we understand and agree with the reviewer’s valuable feedback concerning tracking changes in beliefs over time from the beginning to the end of the course, we cannot, unfortunately, in retrospect gather and add additional data, that is time and process sensitive.

This limitation has now been highlighted in the study’s discussion section, and the edited section now reads:

“Pre-assessments and follow-up work could help to track changes amongst participants and contribute to drawing some conclusions concerning temporal aspects of transformation, environmental engagement, and the impact of education and inner-outer transformation processes over time.”  (p. 15).

Reviewer 2 Report

The approach and aim of the study are very important: to understand the motivational issues in depth in process of change/transition at various levels. I think that the article brings valuable discussion to the theme.

As it could be expected, the participants for this kind of course are already "believers", so I would like to have more detailed description of theirs "profiles": in order to understand values, beliefs, backgrounds behind the comments and results. The limitations related to the participant combination were somehow presented already in the article, but I think it would be important to understand a bit more about this.

Secondly, in order to understand the course "ingredients"/content better, it would be necessary to get more insight, e.g. in form of one or two examples about how different participants "work" during the 1-year course.

Author Response

Manuscript ID challenges-2038792

Response to Reviewers

Reviewer #2

Comments

Responses

1

OVERALL COMMENT & ASSESSMENT

“The approach and aim of the study are very important: to understand the motivational issues in depth in process of change/transition at various levels. I think that the article brings valuable discussion to the theme.”

We thank reviewer two for their positive and helpful feedback concerning the topic and content of our study.

We have aimed to adjust the manuscript in line with the reviewer’s comments.

2

“As it could be expected, the participants for this kind of course are already ‘believers’, so I would like to have more detailed description of their ‘profiles’: in order to understand values, beliefs, backgrounds behind the comments and results. The limitations related to the participant combination were somehow presented already in the article, but I think it would be important to understand a bit more about this.” 

We thank the reviewer for this comment and have added more details regarding participants profile in addition to the self-selecting process of participating in the course. The new text can be found both in the methods-section (p. 6) and the discussion where it is further highlighted in the limitations (p. 16).

3

“Secondly, in order to understand the course ‘ingredients’/content better, it would be necessary to get more insight, e.g. in form of one or two examples about how different participants ‘work’ during the 1-year course.”

We appreciate the reviewer’s questions regarding the program.

In order provide an even better understanding of the course, we have included additional descriptions of the course ‘ingredients’/content with some examples, in the introduction (p. 5).

For additional information, we have previously referred to external sources.

We hope that these clarifications will provide readers more insight into what the program entailed.